# Structural Evolution of Nanophase Separated Block Copolymer Patterns in Supercritical CO_2_

**DOI:** 10.3390/nano11030669

**Published:** 2021-03-08

**Authors:** Tandra Ghoshal, Timothy W. Collins, Subhajit Biswas, Michael A. Morris, Justin D. Holmes

**Affiliations:** 1School of Chemistry, AMBER and CRANN, Trinity College Dublin, D02 AK60 Dublin, Ireland; MORRISM2@tcd.ie; 2School of Chemistry, University College Cork, T12 YN60 Cork, Ireland; collins.incorporated@gmail.com (T.W.C.); s.biswas@ucc.ie (S.B.); j.holmes@ucc.ie (J.D.H.); 3Tyndall National Institute, University College Cork, T12 YN60 Cork, Ireland; 4AMBER Centre, Environmental Research Institute, University College Cork, T23 XE10 Cork, Ireland

**Keywords:** block copolymer, scCO_2_ annealing, microphase separation, structural evolution, feature size variation

## Abstract

Nanopatterns can readily be formed by annealing block copolymers (BCPs) in organic solvents at moderate or high temperatures. However, this approach can be challenging from an environmental and industrial point of view. Herein, we describe a simple and environmentally friendly alternative to achieve periodically ordered nanoscale phase separated BCP structures. Asymmetric polystyrene-*b*-poly(ethylene oxide) (PS-*b*-PEO) thin film patterns of different molecular weight were achieved by annealing in supercritical carbon dioxide (sc-CO_2_). Microphase separation of PS-*b*-PEO (16,000–5000) film patterns were achieved by annealing in scCO_2_ at a relatively low temperature was previously reported by our group. The effects of annealing temperature, time and depressurisation rates for the polymer system were also discussed. In this article, we have expanded this study to create new knowledge on the structural and dimensional evolution of nanohole and line/space surface periodicity of four other different molecular weights PS-*b*-PEO systems. Periodic, well defined, hexagonally ordered films of line and hole patterns were obtained at low CO_2_ temperatures (35–40 °C) and pressures (1200–1300 psi). Further, the changes in morphology, ordering and feature sizes for a new PS-*b*-PEO system (42,000–11,500) are discussed in detail upon changing the scCO_2_ annealing parameters (temperature, film thickness, depressurization rates, etc.). In relation to our previous reports, the broad annealing temperature and depressurisation rate were explored together for different film thicknesses. In addition, the effects of SCF annealing for three other BCP systems (PEO-*b*-PS, PS-*b*-PDMS, PS-*b*-PLA) is also investigated with similar processing conditions. The patterns were also generated on a graphoepitaxial substrate for device application.

## 1. Introduction

Block copolymers (BCPs) are quintessential nanostructure forming materials due to their self-organizing capabilities [1]. This provides a large variety of potential technological applications such as nanolithography and “bottom up” microelectronic device fabrication depending on their different ordered morphologies produced after microphase separation [2,3]. Composition (block ratio, molecular weight, interaction parameters) and temperature are the controlling factors which determine the different morphologies of BCPs [1]. In thin film form, typically obtained by spin-casting or dip-coating procedures, interfacial energies between air and a solid substrate as well as the film thickness relative to the bulk periodicity are additional factors which also determine the morphology of a BCP [3,4,5,6]. Generally, Solvent annealing is utilised to obtain ordered phase-separated, nanostructured thin films at low temperatures and in relatively short time periods through swelling the polymer blocks by a solvent, providing the necessary chain mobility and free volume to the blocks for microphase separation [7,8]. Organic and halogenated solvents are commonly used to swell one or more blocks within a polymer film even though they are toxic and have a negative environmental impact [9]. Removal of excess solvent is also necessary for many technological and biological applications concerning sensitivity [10,11,12,13]. In addition, achieving a stable ordered structure with a controlled orientation throughout a very large area of a substrate is challenging. In contrary, supercritical carbon dioxide (scCO_2_) is an attractive solvent (or diluent) for the phase separation of BCPs as the fluid increases the free volume and chain mobility of dense polymers, thereby reducing the melt viscosity (*T*_g_) and melting point (*T*_m_) and facilitating the formation of self-assembled nanostructures [14,15,16]. Furthermore, the solvation and transport properties of scCO_2_ can be finely adjusted via modest changes in pressure or temperature [17,18,19]. The desirable qualities of CO_2_ include low cost, wide availability, moderate critical conditions (critical temperature (*T*c) = 31 °C, critical pressure (*P*c) = 73.8 bar and critical density (*ρ*c) = 0.468 g.cm^−3^), environmentally and chemically (volatile, inert, non-flammable) benign nature, low interfacial tension and complete elimination of the gas after process termination [20,21,22].

However, phase ordering and kinetics of BCP films by scCO_2_ have been examined from both a theoretical and an experimental perspective, but no clear evidence to support the formation of nanostructured surface periodic arrangements in BCPs annealed at low temperatures in scCO_2_ has been realised [23,24,25]. A few reports on BCP surface structure formation in scCO_2_ have been reported, but this was largely achieved by annealing polymers above their melting points [15,25,26].

Microphase separation of polystyrene-b-polyethylene oxide (PS-*b*-PEO) BCP films into nanostructured arrangement by annealing in scCO_2_ at a relatively low temperature was achieved by our group [27]. The effects of annealing temperature and depressurisation rates for one of the BCP systems were discussed previously. Herein, the structural and dimensional evolution of nanohole and line/space surface periodicity for four different molecular weights PS-*b*-PEO systems, upon changing the scCO_2_ annealing parameters (temperature, pressure, time, film thickness, etc.) is highlighted. Compared to our previous reports, a broad annealing temperature and depressurisation rate ranges were explored to achieve structural evolution along with for different film thicknesses. The effects of BCP nanopatterns for three other BCP systems was also investigated with similar processing conditions. We also discuss the differentiation in the interaction parameters of scCO_2_ with the polymers and thereby the changes in the glass transition temperatures of the BCP films causing microphase separation. The patterns were also generated on a graphoepitaxial substrate for device application.

## 2. Materials and Methods

Polystyrene-*b*-poly(ethylene oxide) (PS-*b*-PEO) diblock copolymers were purchased from Polymer Source Inc., Dorval, QC, Canada and used without further purification. Toluene and acetone were purchased from Sigma-Aldrich, Wicklow, Ireland. Four different molecular weight BCPs were explored with number-average molecular weights of (i) *M*_n_, PS = 9.5 kg mol^−1^, PEO = 5 kg mol^−1^, *M*_w_/*M*_n_ = 1.08; (ii) *M*_n_, PS = 20 kg mol^−1^, PEO = 6.5 kg mol^−1^, *M*_w_/*M*_n_ = 1.05; (iii) *M*_n_, PS = 42 kg mol^−1^, PEO = 11.5 kg mol^−1^, *M*_w_/*M*_n_ = 1.07 and (iv) *M*_n_, PS = 102 kg mol^−1^, PEO = 34 kg mol^−1^, *M*_w_/*M*_n_ = 1.09, where *M*_w_ is the weight-average molecular weight. Single crystal B doped silicon (100) wafers with a native oxide layer were used as substrates. These wafers were cleaned by ultrasonication in acetone and toluene (Sigma-Aldrich, Wicklow, Ireland) for 30 min in each solvent and dried under a nitrogen stream. PS-*b*-PEO was dissolved in toluene to yield a 0.8 wt% polymer solution which was aged for 12 h at room temperature. PS-*b*-PEO thin films were formed by spin coating the polymer solution at different spin speed for 30 s onto a Si wafer. Spin coating speeds were varied to accomplish different film thicknesses. Polymer coated wafer were subsequently loaded into a stainless-steel high-pressure vessel (5 mL, High pressure equipment, PA, USA), which was sealed and placed into a preheated oven (purchased from Thermo Fisher Scientific, UK) at temperatures ranging between 35 and 60 °C and pressurised with CO_2_ at a rate 0.5 mL min^−1^, using a manual pressure generator (ISCO HPLC pump, Teledyne, Lincon, USA). Annealing of the polymer thin films was carried out in scCO_2_ at a constant temperature (oven temperature) and pressure (1000–1400 psi). After the desired time, the annealing chamber was removed and cooled naturally before the CO_2_ pressure was released at a typical depressurisation rate of 40 psi min^−1^ with a pressure controller. Depressurisation was also undertaken at rates of 20, 30, 60 and 120 psi/min to understand its effects on BCP morphologies and the microphase separation process. The microphase separated structures were produced on graphoepitaxial substrate.

Surface morphologies of BCP thin films were imaged by scanning probe microscopy (SPM, Park systems, XE-100, Japan) in tapping mode. The film thicknesses were measured by optical ellipsometry (Woolam M2000, Lincoln, USA) at a fixed angle of 70° for a minimum of five different locations on the sample. Average values were reportesd as the measured thickness value. A two-layer model (SiO_2_ + BCP) was used to simulate experimental data. Samples were prepared for TEM cross sectional imaging with an FEI Helios Nanolab 600i system containing a high resolution Elstar™ Schottky field-emission SEM and a Sidewinder FIB column (purchased from Oregon, USA). A gold protective layer was deposited on top of the film surface before FIB processing. Further platinum e-beam and ion beam was deposited during FIB cross-section. The sectioned Si substrate was transferred to a TEM grid and imaged by transmission electron microscopy (TEM, JEOL 2100, Netherlands).

## 3. Results and Discussion

### 3.1. Dimensional and Structural Control of Block Copolymer Nanopatterns

Dimensional and structural control over the self-assembled block copolymer nanopatterns was achieved using different molecular weight PS-*b*-PEO systems; the compositions of the constituent blocks are represented as S1 (9500–5000), S2 (20,000–6500), S3 (42,000–11,500) and S4 (102,000–34,000). Polymeric solutions were spin coated onto substrates at a spin speed of 3000 rpm for 30 s. The as-coated BCP films formed poorly ordered micelles for S1, S2 compositions whereas S3 exhibited little indication of periodic ordering at room temperature. Poorly ordered microphase separation without any controlled orientation was observed for S4 films. To achieve microphase separation, annealing of the polymer thin films was carried out in scCO_2_ at different temperatures, pressures, holding times and depressurisation rates. Figure 1 shows topographic AFM images of the PS-*b*-PEO systems after annealing in scCO_2_ for 30 min at a temperature of 40 °C. The images show ordered arrangements over large areas for all polymer compositions with no indication of de-wetting. In the AFM images, the PEO cylinders are darker in colour due to its much lower glass transition temperature (*T*_g_) (−60 °C) compared to 105 °C for PS [28,29]. It is believed that the PEO becomes crystallized just after the spin-coating from toluene solution. Additionally, AFM measurements is carried out at room temperature and PEO domains is thought to be softer (semi molten state) than PS regime as reported values of Young’s moduli of PS and PEO were 5.2 and 0.2 GPa, respectively [30]. Therefore, the darker regions must represent the PEO cylinders as the depression of the domains are higher compared to PS during AFM measurement. BCP films with compositions of S1 and S2 were depressurised at a rate of 40 psi min^−1^ to achieve microphase separation and to control the orientation of the cylindrical domains generated. Figure 1a shows a microphase separated arrangement of hexagonal ordered PEO cylinders parallel to the substrate surface for an S1 polymer composition, whereas hexagonally ordered PEO cylinders with a perpendicular orientation were obtained for polymer films with an S2 composition under similar experimental conditions (Figure 1b). The average centre to centre spacings and PEO cylinder diameter were found to be 17 nm and 7 nm, respectively for S1 polymer compositions and 37 nm and 22 nm, respectively for S2 films. For the higher molecular weight polymer systems, i.e., S3 and S4, perpendicular cylinders of PEO cylinders were realised at a scCO_2_ pressure of 1200 psi and a depressurisation rate of 40 psi min^−1^ (Figure 1c,d, respectively). The mean centre to centre spacings were found to be 52 nm and 74 nm for S3 and S4 polymer systems, respectively. The mean PEO cylinder diameter was observed to be 28 and 38 nm for the S3 and S4 films, respectively. At a scCO_2_ pressure of 1400 psi, enhanced surface roughness, thickness variations and pattern degradation (pattern missing) were observed for all of the PS-*b*-PEO systems. The thickness of the films before and after annealing, as determined by optical ellipsometry, remained constant at 19, 36, 52 and 72 nm for S1 to S4 compositions, respectively. Table 1 summarises the SCF annealing conditions, film thicknesses, measured pitch size and PEO cylinder diameters for different PS-*b-*PEO systems. No large scale surface roughness or thickness undulation across the surfaces of the films was noticed and the thickness of the films produced close emulated their domain spacing value. The morphology and interfaces of S3 polymer films was further analysed by cross sectional TEM. A plasma dry etching process was applied to partially remove the PEO domains to increase contrast between the ordered nanofeatures for cross-sectional TEM analysis. The insert of Figure 1c shows the wavy nature of the surface of a film due to partial etching of the PEO domains with a film thickness around 48 nm. The non-selective etching treatment between PS and PEO reduced the film thickness. The arrangement of the PEO domain in the films continues to a definite depth within the film. The mean spacing of features was consistent with a domain spacing of 52 nm as previously measured by AFM. The wave structure was not perfectly periodic due to either/or both the non-selective nature of the etching and/or a non-perfect image projection (since the film may not have been cut exactly orthogonal to the holes). The mean diameter of the PEO domain was observed to be 30 nm, due to broadening during the etching step. The film was strongly adhered to the substrate surface with no indication of deformation or delamination.

Figure 1e shows the variation of pitch size with the total molecular weight and PEO molecular weight for all of the PS-*b*-PEO systems studied. Both of the curves show an increment in pitch size with total and PEO molecular weight as expected.]. An exponential curve fitting for both the curves shows the relationship of pitch size for the BCP.

The variation of pitch size with the total molecular weight can be expressed as
(1)$$y=3.75123+4.56393\;e^{0.04551x}\;$$

The variation of pitch size with the PEO molecular weight can be expressed as
(2)$$y=4.05001+0.25786\;e^{0.06426x}$$

### 3.2. Mechanism of Microphase Separation for Different BCP Systems

Although both the PS and PEO blocks swell in CO_2_, the BCP films returns to their original thickness after complete removal of CO_2_. Microphase separated structures is achieved at relatively lower CO_2_ pressures for all of the molecular weight BCPs, is due to the higher values of *χN* (>10) (*χ* is the Flory–Huggins interaction parameter and *N* is the degree of polymerisation).

The *χ* parameter can be calculated as
(3)$$\chi =\;\frac{V_{p\;\;\;}{\left(\delta_{1\;\;}-\;\delta_{2\;\;}\right)}^{2}\;}{RT}$$
where *V_p_* is the volume of one mole of polymer segments and *δ* is the solubility parameter. The calculated value of *χ* used in this study for the PS-*b*-PEO system is 0.081. The measured *χN* values were 11.21, 20.54, 41.49, 105.42 for S1, S2, S3 and S4, respectively. The higher molecular weight BCP systems requires lower external mediating or a thermodynamic driving force to achieve phase separation compared to lower molecular weight BCPs. This means comparatively lower free energy is required at higher *χN* values to rearrange the polymer chains for achieving self-assembly. The parallel and perpendicular cylinder orientation is dictated by different volume fraction of the constituent blocks. For S1, the volume fraction of PS, f_PS_ is 0.659 and the volume fraction of PEO, f_PEO_ is 0.341. The f_PS_ and f_PEO_ values for S2, S3 and S4 are 0.758, 0.242; 0.788, 0.212; and 0.753, 0.247, respectively. The agreement between volume fraction of minor component PEO over PS and the ratio of the areas of the cylinders and the matrix (from the AFM images) over flat surface, suggest that the ordered cylinders are most likely the PEO domains perpendicular to the surface while the matrix is of PS blocks. Figure 2 shows the calculated phase diagram (*χN* vs. f_PEO_) of the BCP systems belongs to hexagonal cylindrical phase regime (superimposed on the original BCP phase diagram). Table 1 summarises the *χN* values and the volume fractions of the PEO blocks (f_PEO_) for different PS-*b*-PEO systems. The equilibrium stability for vertical orientation is achieved by scCO_2_ annealing for the f_PEO_ values equal to or less than 0.25 for S2, S3 and S4. Vertical orientation represents least entropically hindered route for microphase separation, since the PEO cylinder length is limited to the thickness of the films. Previous studies have shown that the swelling of PEO in scCO_2_ is between 20–25 wt%, larger than that of PS (5–6 wt%) at similar pressure of 1400 psi [27,31,32,33]. Thus, the swelling of the entire film is dominated by the swelling of the PEO microdomains compared to the PS block. The higher volume fraction of the PEO blocks for BCP with an S1 compositions imparts higher mobility to the PEO chains leading to an increment of Gibbs energy within the film due to an increase of the configurational freedom of the PEO chains. This increment in energy within the system easily orientates the cylinders parallel to the substrate which although entropically less favoured. Thus, the PS blocks serves as a majority surface block forming matrix through minimizing the free energy of the system maintaining the equilibrium stability. The exact thickness (relative to cylinder repeat distance) of the monolayer film is also preferred the parallel orientation. However, the selective enhancement of CO_2_ sorption within the PEO blocks compared to the substrate-polymer and polymer-air interfaces overcomes film dewetting. Both the PS and PEO blocks were present at the polymer–air interface after scCO_2_ annealing due to reduced surface tension caused by higher solvent concentration inside the film.

### 3.3. Formation of BCP Nanopatterns at Different SCF Annealing Temperatures

The effects of annealing temperature on the microphase separation and ordered arrangement of the S3 BCP films was investigated as small change in the annealing temperature was found to alter the kinetics (solvation of CO_2_ into the polymer) of the polymer–fluid mixture under the same pressure [34]. Figure 3 shows the topographic AFM images of microphase separated surface structures of S3 polymer compositions annealed between temperatures of 35–60 °C for 30 min at a scCO_2_ pressure of 1200 psi. At a temperature of 35 °C, microphase separated structures with perpendicularly oriented PEO cylinders were formed in most locations on the films (see Figure 3a), but missing patterns (black spots) were also evident. The thickness of the film shown in Figure 3a is slightly higher which was around 55 nm compared to films prepared at 40 °C. At a temperature of 45 °C, phase separated well ordered BCP film with two different thicknesses of around 52 nm and 55 nm were formed (Figure 3b). Thicker fragments had larger cylinder diameters than thinner regions. Increasing the temperature to 50 and 60 °C, resulted in the formation of short parallel oriented PEO cylinders as well as perpendicularly orientated regions. Moreover, the PEO cylinders are with larger diameter at 50 °C (compared to lower temperature) whereas few of them flipped their orientation to parallel to the substrate (Figure 3c). More flipping tendency (to parallel) is noticed at the highest temperature of 60 °C (Figure 3d). The increased disparity between the CO_2_–PS and CO_2_–PEO solubility and interfacial energies at high temperatures creates larger effective interaction parameter (*χ*_eff_) irrespective of the film thickness, eases the process of phase separation. Previous reports suggests that the temperatures below 50 °C are more suitable to achieve phase separation under constant CO_2_ pressure conditions, consistent with our finding [27,32]. At a temperature of 60 °C, which is just above the melting temperature of the PEO block (55 °C) for the S3 molecular weight system, PEO is in a semi-molten state which facilitates the reorientation of the PEO cylinders and the morphological change [35]. Of note, the larger free energy at higher temperature can cause frequent structural transition in order to release the extra tension. The process of the forming nanopatterns by scCO_2_ annealing is relatively faster than that was reported for solvent (organic) annealing [29,36]. This is due to the influential reduction in the viscosity of PEO chains by scCO_2_ diffusion into the block which reaches to a critical value, thus permitting the polymer chains easily extend and/or relax to achieve phase separation. At high temperatures, the PEO chains extends and swell more compared as that of lower temperatures, further rapid removal of scCO_2_ during depressurisation shrinks back the film quickly leads to island formation of different thicknesses.

### 3.4. Effects of Film Thicknesses by Varying Spin Coating Speed

As discussed previously the change in film thickness during the self-assembly process significantly influences the final morphology, size and ordering of the BCP films. Figure 4 shows AFM images of BCP films with an S3 composition with varying spin coating speed after microphase separation, achieved by annealing in scCO_2_ at a temperature of 40 °C and a pressure of 1200 psi for 30 min. At all spin speeds between 1000–4000 rpm, microphase separated perpendicularly oriented PEO cylinders were formed inside the PS matrix. Variations in the thickness of the films, in the form of island was evident, throughout the films at all spin speeds investigated. The islands (black contrast) are thinner than the regular film (white) regions. Moreover, the frequency and size of the islands increased with increasing spin speed. At a low spin speed of 1000 rpm, the PEO cylinder diameters were ~25 nm in most of the film areas, whereas the black contrast islands had larger diameters around 27 nm (Figure 4a). Increasing the spin speed to 2000 rpm, resulted in the formation of PEO cylinders in white film regions with a mean diameter of 27 nm compared to those in the black regions where the mean diameter was 29 nm (Figure 4b). At spin speeds of 3000 rpm, the mean diameters of the PEO cylinders in the white and dark regions were both found to be 28 nm. Further increasing the spin speed to 4000 rpm alters the mean cylinder diameter to 30 and 26 nm in the black and white film regions, respectively. (Figure 4c). The measured thickness of the films before and after phase separation were 59 nm, 55 nm and 50 nm, for spin speeds of 1000, 2000 and 4000 rpm, respectively. The results are summarised in Table 2. The differences in film thicknesses and uniformity in PEO cylinder diameter after microphase separation with the varying spin speed can be explained based on the trapped scCO_2_ during annealing. For different thickness of the films, the kinetics are different due to the vitrification of the film that alters the swelling behaviour of the blocks. The trapped solvent results in defects in the surface morphology by modification in the polymer-substrate, polymer-air and polymer-polymer interactions. The increased ordering and the uniformity of patterns for the thinnest films could be due to significantly less residual solvent and lack of a well-defined solvent front. For thicker films, longer time is required to evaporate the solvent due to combined effects of sufficiently large film thickness and attractive interaction of the solvent with the substrate [37]. The presence of significant amount of trapped solvent in thicker films may help to form the bigger island type structure due to two-dimensional Ostwald ripening type mechanisms [28]. The PEO cylinders swells and shrinks back faster through the release of trapped solvent for the islands with black contrast (thinner regions) than the thicker islands. Thus, thinner islands are with less cylinder diameter than white regions (more trapped solvents for thicker part) due to higher amount of PEO block swelling.

### 3.5. Effects of Depressurisation Rate on the Self Assembly

As previously reported, annealing BCPs in scCO_2_ effectively decreases their glass transition temperature (*T*_g_), which plays an important role in their microphase separation and the creation of ordered patterns [17]. *T*_g_ of PEO is always below room temperature and *T*_g_ of PS is reported to be around 30 °C in scCO_2_ [38]. At high depressurization rate, such as 120 psi/min, depressions or holes on the surface of a films are readily formed [15]. Thus, it is convincing to study the effects of depressurisation rates on the internal structural transition as well as surface morphology of the BCP films. Figure 5 shows the surface and internal morphologies BCP films with an S3 composition after annealing in CO_2_ at a CO_2_ pressure of 1200 psi and a temperature of 40 °C for 30 min, with depressurization rates of 120, 60, 30 and 20 psi/min. At depressurisation rates of 120 and 60 psi/min, wafer scale BCP films with hexagonally ordered PEO cylinders containing holes (dark spots in the AFM image) throughout their surface were observed. In addition, the non-uniform diameter of the PEO cylinders formed and thickness variation in the films were also noticed (Figure 5a,b). These holes arise from the rapid depressurisation of the CO_2_ which forces its way back to the polymer-air interface during depressurisation. For cross-sectional TEM imaging, samples were partially treated with ethanol to etch and/or modify the PEO domains in order to increase the electron contrast and highlights the surface and internal morphology of the film (Figure 5c). At the surface of the film, undulating patterns of non-uniform PEO cylinders (dark contrast) was observed. Within and throughout the interior of the film, a well-resolved array of equally spaced ordered pores or lines was observed consistent with the expected etch/modification of the PEO domain by the ethanol treatment. The enhancement of the contrast can be associated with the partial crystallisation of the domains as crystalline PEO has much greater density (1.24 g cm^−3^) than amorphous PEO (1.12 g cm^−3^) [28]. The PEO cylinder diameter varied between 27–33 nm. Multiple layers of voids or spaces can be seen across the film thickness as the film thickness increases to 72 nm. Continuous parallel layers of voids or elliptical ordered patterns is noticed depending on the orientation of the PEO cylinders. The void spaces formed during fast depressurisation through the rapid removal of CO_2_ from the swelled PEO sites providing additional thickness to the film. Another reason for the void generation could be due to a sudden increase in the viscosity of the polymers during fast depressurisation. The optimum depressurisation rate of 40 psi/min was found to produce uniform diameter PEO cylinders in hexagonal arrangements. By decreasing the depressurisation rate to 30 psi/min, a narrow range of PEO cylinder diameter distribution observed but the film was distorted in several places (Figure 5d). Thus, discontinuous film with thickness around 50 nm were noted possibly due to high local concentration of CO_2_ during degassing resulted in the collapse of the ordered surface structure of the BCP films. Further decreasing the depressurisation rate to 20 psi/min, thickness variations with mixed PEO cylinder orientation (both parallel and perpendicular) was observed throughout the film (Figure 5e). This observation is probably related to a large amount of trapped CO_2_ causing further film swelling. The flipping tendency is due to minimise the Gibbs energy of the system [5].

### 3.6. Fabrication of BCP Nanopatterns on Graphoepitaxial Substrate

In this study, we also examine whether the scCO_2_ annealing process could be used to achieve ordered BCP films on graphoepitaxial substrates, with different channels widths. A graphoepitaxial substrate was used to generate ordered microphase separated PS-*b*-PEO, S3 BCP nanopatterns. A 7 nm thick silica layer coated Si substrate with 50 nm deep topographically defined patterns of SiN sidewall was used as a substrate. The concentrations of the BCP-toluene solution for spin coating were calibrated to 0.5 wt% to avoid any overfilling within the channels. Figure 6a shows ordered arrays of dot patterns within channel widths of 120 and 240 nm after annealing in scCO_2_ at a temperature of 40 °C and a pressure of 1200 psi for 30 min. Similarly ordered arrays were also realised for 160 and 320 nm channel widths as shown in Figure 6b,c, respectively. All the images reveal that the thickness variation increases and the ordered arrangement decreases with increasing channel widths. However, the BCP film all segregate into regular domain spacings of around 50 nm with mean PEO cylinder diameter of 27 nm. This result indicates that the scCO_2_ annealing process is also applicable to use with graphoepitaxy-defined substrates.

### 3.7. Exploring SCF Annealing for Other BCP Composition

The efficiency of the scCO_2_ annealing was also investigated for other BCP systems. In this context, an inverse PS-*b*-PEO (16,000–39,500), Polystyrene-*b*-polymethylsiloxane (PS-*b*-PDMS) (31,000–11,000) and Polystyrene-*b*-polylactic acid (PS-*b*-PLA) (21,000–9000) BCPs were annealed in scCO_2_ for 30 min at a temperature of 40 °C with a pressure of 1200 psi. The AFM images shown in Figure 7 revealed the ordered arrangements for all of the BCPs. Figure 7a shows parallel orientation of PS cylinders inside PEO matrix with an average cylinder to cylinders spacing of 50 nm and cylinder diameter of 20 nm. The self-assembly was realised at a lower temperature (40 °C) compared to those achieved by solvent annealing process (60 °C) [39]. A mixed orientation (parallel and perpendicular to the substrate surface) of PDMS cylinders inside the PS matrix is observed for scCO_2_ annealing with similar conditions (Figure 7b). The average cylinder to cylinders spacing is 44 nm and the cylinder diameter is 18 nm. In comparison with the observation by Rasappa et al. [40] for the self-assembly achieved by solvent annealing, the average cylinder to cylinders spacing is slightly smaller compared to our study. Figure 7a,b shows an enhanced surface roughness for PS-*b*-PDMS compared to the PS-*b*-PEO systems. In case of PS-*b*-PLA, similar scCO_2_ annealing parameters provides a hexagonal ordered perpendicularly oriented PLA cylinders inside PS matrix with a smooth surface (Figure 7c). The average cylinder to cylinders spacing is 34 nm whereas the cylinder diameter is 12 nm. Generally, microphase separation for both of the cylinder forming and lamellar forming PS-PLA BCPs were realised by solvothermal annealing approach. For both of the systems, the periodicity between the blocks was observed smaller than (~5 nm) that of observed in this study by SCF method [41,42]. This tendency might corresponds to more swelling tendency of PLA and PDMS by SCF compared to organic solvents. These results also indicate the efficiency of the scCO_2_ annealing to achieve ordered self-assembled patterns for a range of BCP systems.

## 4. Conclusions

Self-assembled periodically ordered PS-*b*-PEO nanopatterns in thin films have been realised by annealing in scCO_2_ at low temperatures and pressure. A broad range of dimensional variation, i.e., mean centre-to-centre cylinder spacings (between 17–72 nm) and mean PEO cylinder diameters (7–38 nm) was achieved using different asymmetric molecular weight BCP systems. Significantly, a morphological evolution with parallel and/or perpendicular orientation of the PEO cylinders was also observed for different BCP molecular weight systems without disrupting the long-range order. High molecular weight systems required low scCO_2_ pressures to achieve microphase separated nanostructures compared to lower molecular weight BCP systems. Lower annealing temperature leads to discontinuous patterns including formation of holes in distinct places whereas higher temperature resulted in pattern degradation and surface roughening in the form of islands simultaneously varying PEO cylinder diameter. Similar structural variations were observed by varying film casting spin speeds. CO_2_ predominantly interacts with the PEO blocks compared to PS imparting higher mobility and surface energy, responsible for the microphase separation. The effective increase in the interaction parameter and decrease of the glass transition temperatures of the BCP films under scCO_2_, compared to vacuum, also leads to phase separation of the BCP thin films. Nanopores or nanocellular structures could be introduced into the films with a periodic surface arrangement by varying the CO_2_ depressurisation rate without the use of an additional solvent. The trapped residual solvent leads to morphological transition in the film structure. Complete elimination of CO_2_ at the end of the process makes the films useful for many technological and biological applications. This unique approach can also potentially be used to form highly ordered nanostructured surface arrangements on the graphoepitaxial substrate for nanolectronic devices. The efficiency of scCO_2_ annealing is established to generate ordered self-assembled patterns for a range of BCP systems.

## Figures and Tables

**Figure 1 nanomaterials-11-00669-f001:**
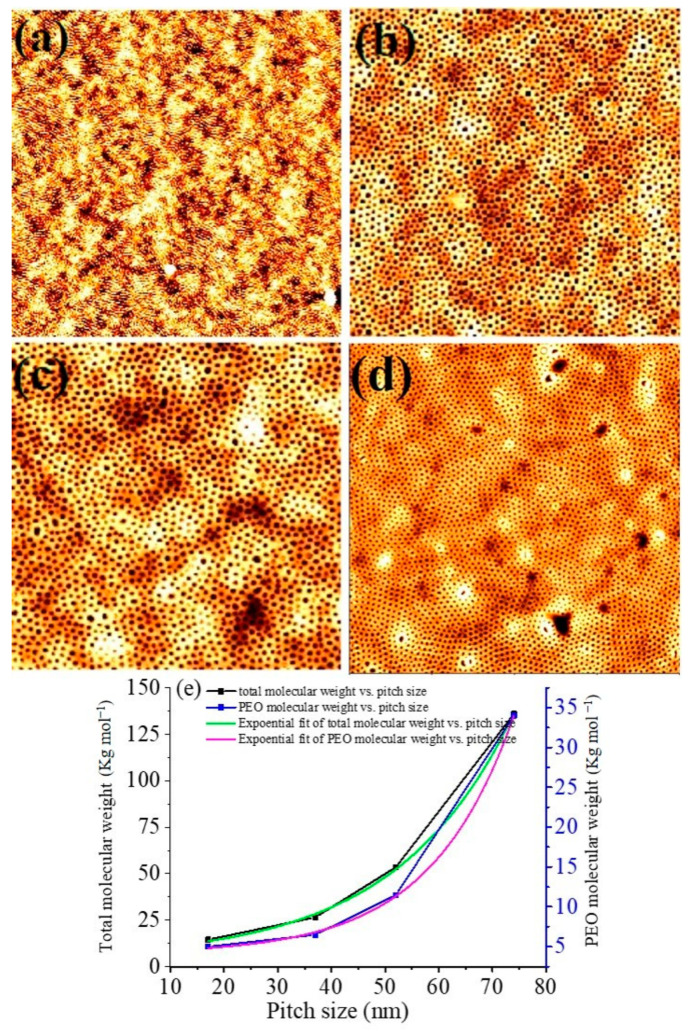
AFM topographic images of microphase separated PS-*b*-PEO thin films patterns under SCF annealing at 40 °C for 30 min of different molecular weights of (**a**) 9500–5000, (**b**) 20,000–6500, (**c**) 42,000–11,500 and (**d**) 102,000–34,000. Scale bar: (**a**–**c**) 2 µm^2^ and (**d**) 4 µm^2^. Inset of (**c**) shows corresponding FIB thinned TEM image. (**e**) pitch size dependence with the total and PEO molecular weights for PS-PEO.

**Figure 2 nanomaterials-11-00669-f002:**
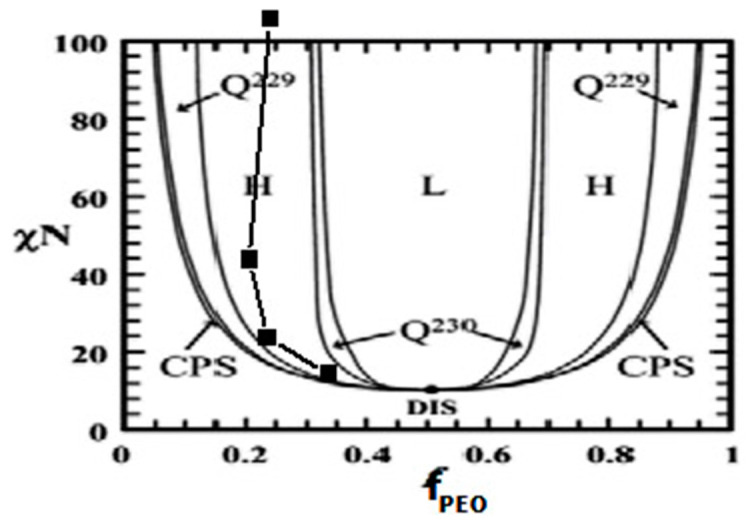
The calculated phase diagram (*χN* vs. f_PEO_) of different PS-*b*-PEO BCP systems (superimposed on the original BCP phase diagram).

**Figure 3 nanomaterials-11-00669-f003:**
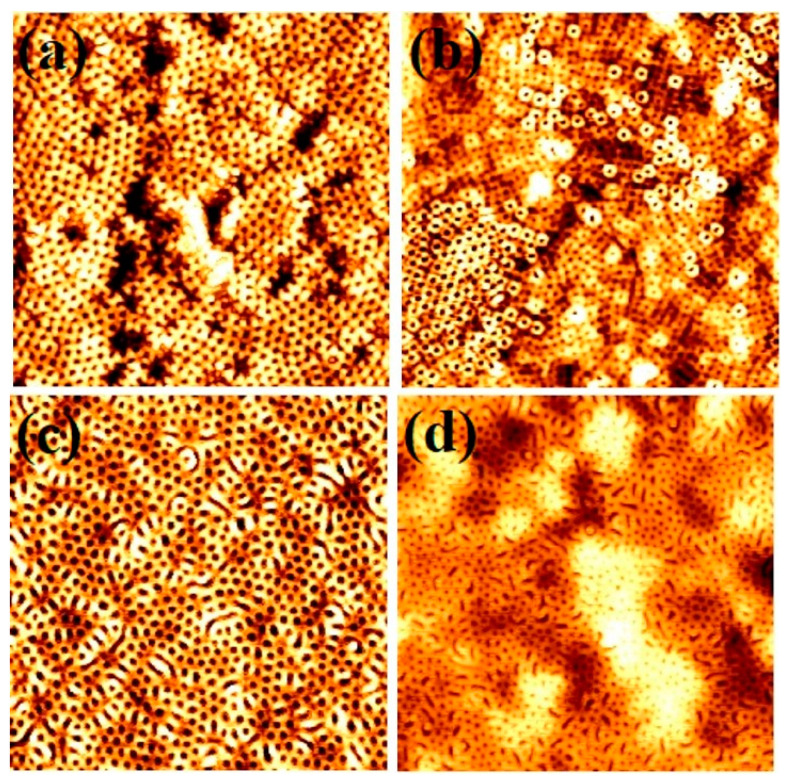
AFM topographic images of microphase separated 42,000–11,500 PS-*b*-PEO thin films patterns under SCF annealing for 30 min formed at different temperature of (**a**) 35 °C, (**b**) 45 °C, (**c**) 50 °C and (**d**) 60 °C. Scale bar: 2 µm^2^.

**Figure 4 nanomaterials-11-00669-f004:**
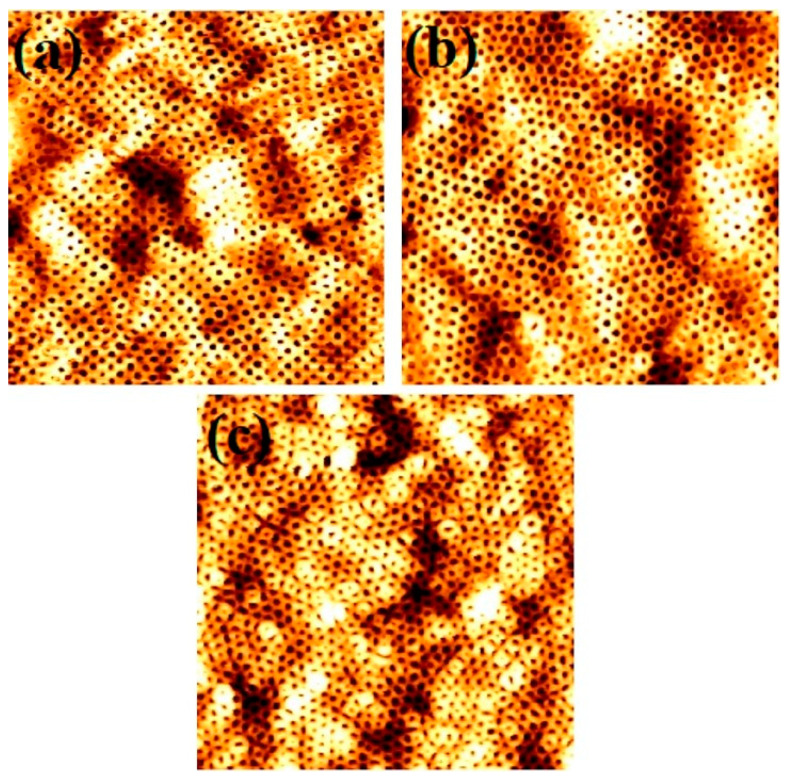
AFM topographic images of microphase separated 42,000–11,500 PS-b-PEO thin films patterns under SCF annealing at 40 °C for 30 min formed by spin coating with different spin speed of (**a**) 1000 rpm, (**b**) 2000 rpm and (**c**) 4000 rpm. Scale bar: 2 µm^2^.

**Figure 5 nanomaterials-11-00669-f005:**
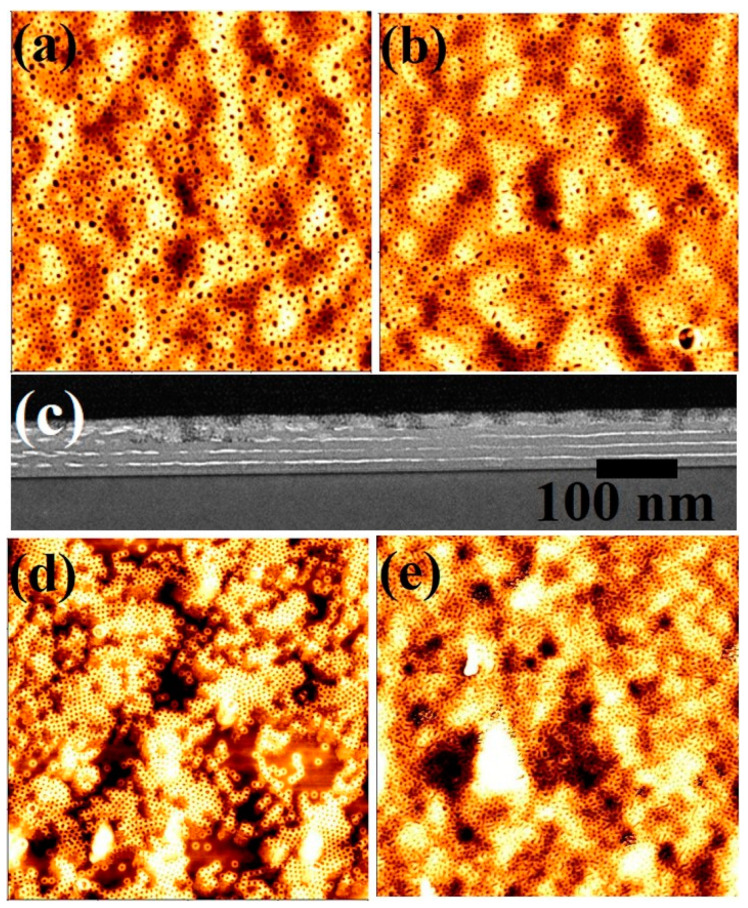
AFM topographic images of microphase separated 42,000–11,500 k PS-*b*-PEO thin films patterns under SCF annealing at 40 °C for 30 min with different depressurization rate of (**a**) 120 psi/min, (**b**) 60 psi/min (**d**) 30 psi/min and (**e**) 20 psi/min. Scale bar: 4 µm^2^. (**c**) FIB thinned cross-sectional TEM image of S3 after SCF annealing at 40 °C for 30 min with a depressurization rate of 120 psi/min.

**Figure 6 nanomaterials-11-00669-f006:**
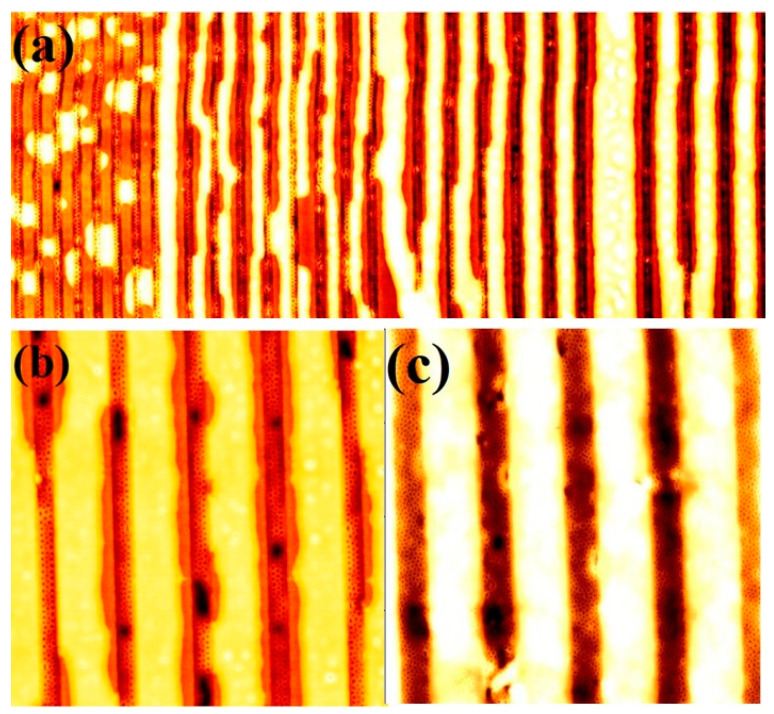
42,000–11,500 PS-*b*-PEO thin film patterns on a graphoepitaxy substrate of different channel width under SCF annealing at 40 °C for 30 min. Scale bar: (**a**) 5 µm × 12 µm, (**b**,**c**) 4 µm^2^.

**Figure 7 nanomaterials-11-00669-f007:**
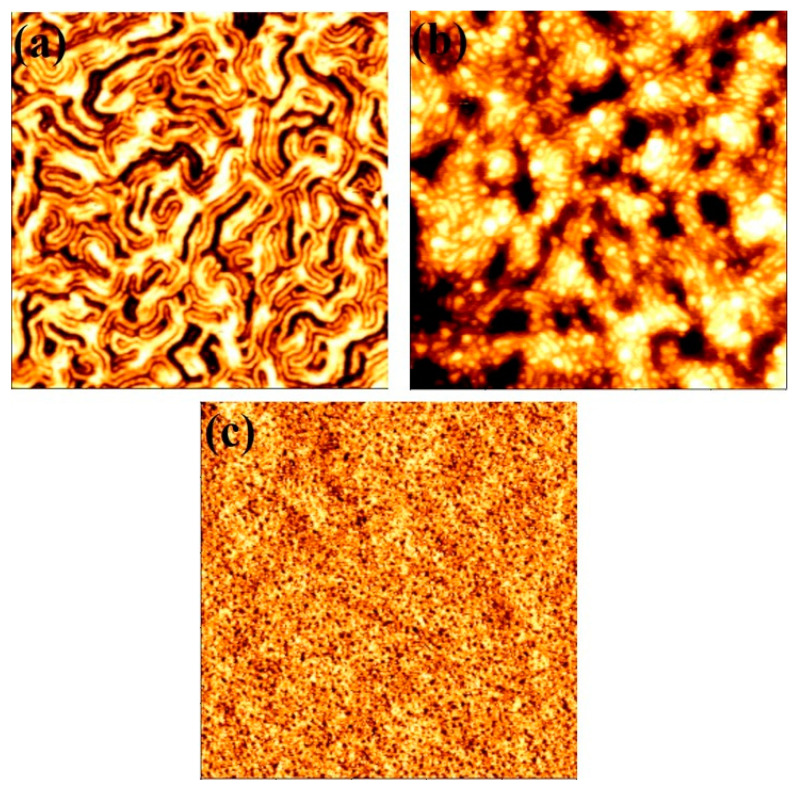
AFM topographic images of microphase separated thin films patterns under SCF annealing at 40 °C for 30 min at a pressure of 1200 psi for a range of BCP systems. (**a**) Inverse PS-*b*-PEO (16,000–39,500), (**b**) Polystyrene-b-polymethylsiloxane (PS-b-PDMS) (31,000–11,000) and (**c**) Polystyrene-b-polylactic acid (PS-b-PLA) (21,000–9000). Scale bar: 2 µm^2^.

**Table 1 nanomaterials-11-00669-t001:** Dimensional and structural control of block copolymer nanopatterns for different PS-*b*-PEO thin films systems under SCF annealing at 40 °C for 30 min.

PS-b-PEO (Molecular Weight)	*χN*	f_PEO_	SCF Annealing Pressure (psi)	Film Thickness (nm)	Pitch Size (nm)	PEO Cylinder Diameter (nm)
9500–5000	11.21	0.341	1300	19	17	7
20,000–6500	20.54	0.242	1300	36	37	22
42,000–11,500	41.49	0.212	1200	52	52	28
102,000–34,000	105.42	0.247	1200	72	74	38

**Table 2 nanomaterials-11-00669-t002:** Effects of spin casting speed for S3 BCP system.

Spin Coating Speed (rpm)	Film Thickness (nm)	PEO Cylinder Diameter (nm) in Black Region	PEO Cylinder Diameter (nm) in White Region
1000	59	27	25
2000	55	29	27
3000	52	28	28
4000	50	30	26

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
