# Peer review of "Structural Evolution of Nanophase Separated Block Copolymer Patterns in Supercritical CO_2"

_nanomaterials, 2021, doi:10.3390/nano11030669_

Round 1
Reviewer 1 Report
The manuscript "Structural Evolution of Nanophase Separated Block Copolymer Patterns in Supercritical CO2 " by Holmes and coworkers deals with the fabrication of block copolymer films and their annealing with scCO2. The manuscript shows interesting research and should be published after some revisions.
General comments:
1. I would suggest to add subsections to make the reading more convenient.
2. How do the formed phases fit into the phase diagram (compared to literature)?
3. In my opinion the manuscript should also feature some kind of summary Scheme or Table to visualize the results in an easy way if possible.
4. In Figure 5, the widths of the substrates should be mentioned in the caption.
5. I think the effect of molecular weight on spacings should be discussed more thoroughly to show the correlation.
6. How do the observed spacings for the other block copolymers (PLA, PDMS) fit to the expectations for such systems? I think a better comparison with traditional methods should be discussed.
Specific comments to check:
7. Abstract: The sentence starting with "Compared" should be reworded
8. Intro: The sentence starting with "Although phase" should be reworded
9. Experimental: check "fixed angle of 70°C"
10. Results: The sentence starting with "Thus, it is of significant" should be reworded
11. Results: The sentence starting with "This results highlights that the" should be reworded
12. The whole manuscript: "b" in "-b-" should be italic
13. The whole manuscript: the abbreviation scCO2 is inconsistent
14. The whole manuscript: check for the space between number and °C
Author Response
Answer to reviewers:
Reviewer 1:
The manuscript "Structural Evolution of Nanophase Separated Block Copolymer Patterns in Supercritical CO2" by Holmes and coworkers deals with the fabrication of block copolymer films and their annealing with scCO2. The manuscript shows interesting research and should be published after
Comment 1. I would suggest to add subsections to make the reading more convenient.
Answer: We have added subsections in the ‘results and discussion’ section in the revised manuscript.
Comment 2. How do the formed phases fit into the phase diagram (compared to literature)?
Answer: We have included the calculated phase diagram (χN vs. fPEO) of the PS-b-PEO BCP systems which belongs to hexagonal cylindrical phase regime and superimposed with the original BCP phase diagram as shown in Figure 2 in the revised manuscript. The corresponding description is also added in the relevant paragraph.
Comment 3. In my opinion the manuscript should also feature some kind of summary Scheme or Table to visualize the results in an easy way if possible.
Answer: We have summarised few of the results and calculations in Table format as Table 1 and 2 in the revised manuscript.
Comment 4. In Figure 5, the widths of the substrates should be mentioned in the caption.
Answer: It was already described in the original manuscript as (a) 5 x 12 µm, (b and c) 4 x 4 µm.
Comment 5. I think the effect of molecular weight on spacings should be discussed more thoroughly to show the correlation.
Answer: We have included a graph (Figure 1e) showing the variation of pitch size with the total molecular weight and PEO molecular weight for all of the PS-b-PEO systems studied in the revised manuscript. The description and curve fitting etc. is added in the relevant paragraph.
Comment 6. How do the observed spacings for the other block copolymers (PLA, PDMS) fit to the expectations for such systems? I think a better comparison with traditional methods should be discussed.
Answer: We have compared and discussed the spacing for the other block copolymers (PLA, PDMS) and added in the relevant paragraph in the revised manuscript.
Specific comments to check:
Comment 7. Abstract: The sentence starting with "Compared" should be reworded.
Answer: This is reworded in the revised manuscript.
Comment 8. Intro: The sentence starting with "Although phase" should be reworded.
Answer: This is reworded in the revised manuscript.
Comment 9. Experimental: check "fixed angle of 70°C".
Answer: We have changed this to 70 ° in the revised manuscript.
Comment 10. Results: The sentence starting with "Thus, it is of significant" should be reworded.
Answer: This is reworded in the revised manuscript.
Comment 11. Results: The sentence starting with "This results highlights that the" should be reworded.
Answer: This is reworded in the revised manuscript.
- The whole manuscript: "b" in "-b-" should be italic.
Answer: We have checked carefully and all ’b’ is in italics in the revised manuscript.
Comment 13. The whole manuscript: the abbreviation scCO2 is inconsistent.
Answer: We have checked carefully and changed all to scCO2 in the revised manuscript.
Comment 14. The whole manuscript: check for the space between number and °C.
Answer: We have checked carefully and changed in the revised manuscript.
Reviewer 2 Report
The manuscript describes nanoscale phase separation of selected block copolymers (commercially available) by annealing in scCO2. An effect of broad annealing temperature and depressurization rate with different polymeric film thicknesses on the patterns obtained was studied. Overall the work is interesting and could be published after minor revision. Here are my remarks/questions:
1. Could you please explain briefly connection between “color” of the AFM images and Tg - chemical composition of the observed domain (p. 3).
2. How did you determine Flory–Huggins interaction parameter (p. 4)?
Author Response
Answer to reviewers:
Reviewer 2:
The manuscript describes nanoscale phase separation of selected block copolymers (commercially available) by annealing in scCO2. An effect of broad annealing temperature and depressurization rate with different polymeric film thicknesses on the patterns obtained was studied. Overall the work is interesting and could be published after minor revision. Here are my remarks/questions:
Comment 1. Could you please explain briefly connection between “color” of the AFM images and Tg - chemical composition of the observed domain (p. 3).
Answer: The explanation is included in the revised manuscript.
Comment 2. How did you determine Flory–Huggins interaction parameter (p. 4)?
Answer: The calculation of Flory–Huggins interaction parameter and explanation is included in the revised manuscript.
